# Cryo-electron tomography reveals the microtubule-bound form of inactive LRRK2

Siyu Chen[1,2,3], Tamar Basiashvili[1,2], Joshua Hutchings[1], Marta Sanz Murillo[2,4], Amalia Villagran Suarez[2,4], Erica Xiong[2,4], Jaime Alegrio Louro[2,4], Andres E Leschziner[1,2,4]*, Elizabeth Villa[1,2,3]*

[1]Department of Molecular Biology, University of California, San Diego, San Diego, United States; [2]Aligning Science Across Parkinson's (ASAP) Collaborative Research Network, Chevy Chase, United States; [3]Howard Hughes Medical Institute, Chevy Chase, United States; [4]Department of Cellular and Molecular Medicine, School of Medicine, University of California, San Diego, San Diego, United States

## eLife Assessment

In this manuscript, Chen et al. used cryo-ET and in vitro reconstituted system to demonstrate that the autoinhibited form of LRRK2 can also assemble into filaments on the microtubule surface, with a new interface involving the N-terminal repeats that were disordered in the previous active-LRRK2 filament structure. The structure obtained in this study is the highest resolution of LRRK2 filaments done by subtomogram averaging, representing a major technical advance compared to the previous paper from the same group. This is an **important** study, especially considering the pharmacological implications of the effect of inhibitors of the protein. The strengths of the data are **convincing**, but the study would be considerably strengthened if the authors explored the physiological significance of the new interfaces and the **incomplete** decoration of microtubules described here.

*For correspondence:
aleschziner@health.ucsd.edu (AEL);
evilla@ucsd.edu (EV)

**Competing interest:** The authors declare that no competing interests exist.

**Abstract** Parkinson's disease (PD) is the second most common neurodegenerative disorder. Mutations in human leucine-rich repeat kinase 2 (LRRK2), a multi-domain protein containing both a kinase and a GTPase, are a leading cause of the familial form of PD. Pathogenic LRRK2 mutations increase LRRK2 kinase activity. While the bulk of LRRK2 is found in the cytosol, the protein associates with membranes where its Rab GTPase substrates are found, and under certain conditions, with microtubules. Integrative structural studies using single-particle cryo-electron microscopy and in situ cryo-electron tomography (cryo-ET) have revealed the architecture of microtubule-associated LRRK2 filaments, and that formation of these filaments requires LRRK2's kinase to be in the active-like conformation. However, whether LRRK2 can interact with and form filaments on microtubules in its autoinhibited state, where the kinase domain is in the inactive conformation and the N-terminal LRR domain covers the kinase active site, was not known. Using cryo-ET, we show that full-length human LRRK2 can oligomerize on microtubules in its autoinhibited state. Both WT-LRRK2 and PD-linked LRRK2 mutants formed filaments on microtubules. While these filaments are stabilized by the same interfaces seen in the active-LRRK2 filaments, we observed a new interface involving the N-terminal repeats that were disordered in the active-LRRK2 filaments. The helical parameters of the autoinhibited-LRRK2 filaments are different from those reported for the active-LRRK2 filaments. Finally, the autoinhibited-LRRK2 filaments are shorter and less regular, suggesting they are less stable.

## Introduction

Parkinson's disease (PD) is one of the most common neurodegenerative disorders, with about 10% of cases being familial (*Monfrini and Di Fonzo, 2017*). A frequent cause of familial PD is mutations in leucine-rich repeat kinase 2 (LRRK2) (*Zimprich et al., 2004*; *Simón-Sánchez et al., 2009*), a multi-domain protein containing both a kinase and a GTPase. Common pathogenic LRRK2 mutations include R1441C/G/H, Y1699C, G2019S, and I2020T (*Simón-Sánchez et al., 2009*; *Satake et al., 2009*). These mutations are autosomal gain-of-function, increasing the kinase activity of LRRK2 (*West et al., 2005*; *Gloeckner et al., 2006*). Genetic studies have also revealed links between LRRK2 mutations and sporadic cases of PD (*Simón-Sánchez et al., 2009*; *Satake et al., 2009*; *Rocha et al., 2022*), and increased kinase activity in an otherwise wild-type LRRK2 in postmortem samples from idiopathic PD patients has also been reported (*Di Maio et al., 2018*). These observations have made LRRK's kinase a major target for PD therapeutics (*Rocha et al., 2022*; *Tang et al., 2023*; *Hu et al., 2023*).

LRRK2 is a 2527-residue, 286 kDa protein. Its N-terminal half is composed of armadillo (ARM), ankyrin (ANK), and leucine-rich repeat (LRR) domains. The C-terminal half contains a Ras-like guanosine triphosphatase (GTPase, or Ras of complex, ROC) domain, followed by a 'C-terminal of ROC' (COR) domain, a kinase domain, and a WD40 domain, with the latter found at the C terminus (*Figure 1A*). Structures of both LRRK2's C-terminal catalytic half (LRRK2[RCKW]) and full-length LRRK2 have been solved by cryo-electron microscopy (cryo-EM), along and bound to both nucleotides and kinase inhibitors (*Deniston et al., 2020*; *Myasnikov et al., 2021*; *Zhu et al., 2024*; *Sanz Murillo et al., 2023*). Wild-type and PD-mutant LRRK2 show a similar overall architecture, with homodimerization mediated by a COR:COR interface (*Myasnikov et al., 2021*; *Zhu et al., 2023*).

Small-molecule LRRK2 kinase inhibitors have been used both for biological studies and in clinical trials (*Tang et al., 2023*; *Hu et al., 2023*). In general, type-I inhibitors capture kinases in a closed conformation by binding to the ATP-binding site and target the kinase in its 'active-like' conformation, inhibiting its kinase activity. Meanwhile, type-II inhibitors capture them in an open, inactive conformation. MLi-2 is a widely used LRRK2-specific type-I kinase inhibitor with very high potency (*Fell et al., 2015*; *Scott et al., 2017*). Although no LRRK2-specific type-II inhibitors have been reported, broad spectrum type-II inhibitors, such as GZD-824 (*Ren et al., 2013*), have been shown to bind LRRK2 with high affinity (*Deniston et al., 2020*; *Sanz Murillo et al., 2023*). As expected, the type-I inhibitor MLi-2 locks the LRRK2[RCKW] kinase domain in the closed, active-like conformation (*Deniston et al., 2020*; *Sanz Murillo et al., 2023*; *Snead et al., 2022*), while the type-II inhibitor GZD-824 stabilizes the kinase domain in the open, inactive conformation (*Sanz Murillo et al., 2023*). Interestingly, recent high-resolution structures of full-length, in-solution LRRK2 show its kinase in an open conformation regardless of which kinase inhibitor was bound (*Sanz Murillo et al., 2023*). In that study, we suggested that the conformational restraining imposed by the N-terminal repeats of full-length LRRK2 prevent the kinase domain from closing. Thus, LRRK2 autoinhibition results from the combination of this effect and a physical blocking of the kinase's active site by the LRR domain (*Sanz Murillo et al., 2023*; *Schmidt et al., 2021*).

When overexpressed, most PD-mutant hyperactive LRRK2 variants show strong co-localization with microtubules (*Kett et al., 2012*). One such mutant, LRRK2[I2020T], has been used to solve its in situ structure on microtubules using cryo-ET (*Watanabe et al., 2020*). In vitro reconstitution of microtubules with LRRK2[RCKW] also revealed a similar filamentous assembly on microtubules (*Snead et al., 2022*). LRRK2 inhibitors, such as MLi-2 and LRRK2-IN-1, augment LRRK2's co-localization with microtubules (*Deniston et al., 2020*; *Schmidt et al., 2019*; *Blanca Ramírez et al., 2017*). MLi-2 stabilizes this interaction and was used for the cryo-EM reconstruction of the in vitro reconstituted microtubules-associated LRRK2[RCKW] filaments (*Snead et al., 2022*). On the other hand, it has been shown that type-II inhibitors decrease the formation of LRRK2 filaments in cells, even for hyperactive PD mutations (*Deniston et al., 2020*; *Schmidt et al., 2021*). Whether binding of LRRK2 to microtubules occurs under physiological conditions, if and how this is affected by PD-linked mutations, and how this could affect intracellular trafficking remain open questions.

The same COR:COR interface identified in the LRRK2 homodimer is present in the microtubule-associated LRRK2 (in situ) and LRRK2[RCKW] (in vitro reconstituted) filaments. In the structure of microtubule-bound filaments in situ, the N-terminal half of full-length LRRK2[I2020T] was not resolved during subtomogram analysis, suggesting that this part of the protein was

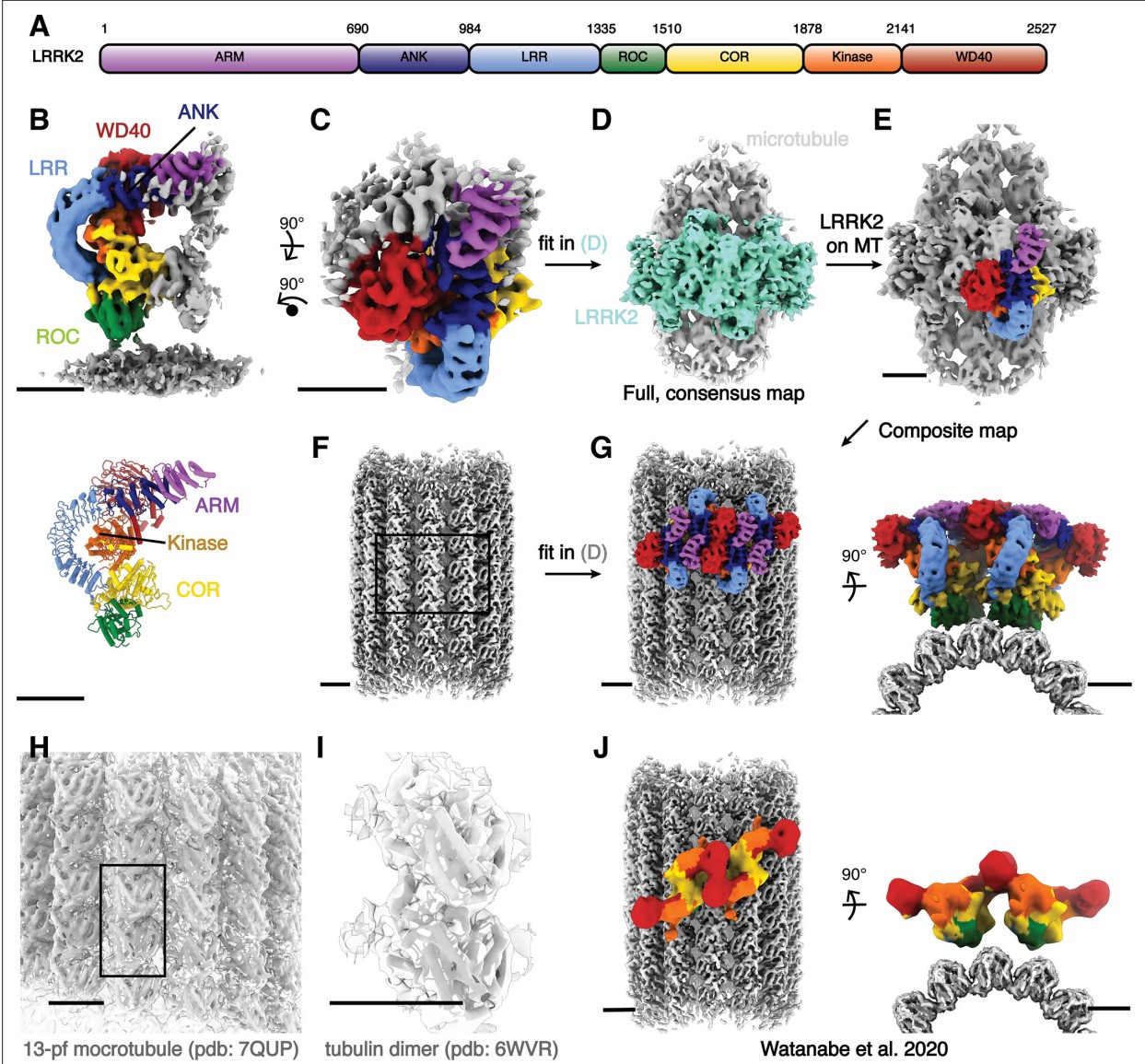

**Figure 1.** Cryo-electron tomography (cryo-ET) structure of the microtubule-associated autoinhibited LRRK2$^{I2020T}$. (**A**) Domain organization of LRRK2. The color coding of domains is used throughout this work. (**B**) Front view of the focused-refined cryo-ET map (left, see Methods) of microtubule-bound LRRK2$^{I2020T}$ assembled in the presence of MLi-2 (dataset (2)), and the model (right) generated from the map. (**C–G**) Graphical representation of the composite LRRK2–microtubule map building process. The full, low-resolution microtubule–LRRK2 map shown in (**D**) was used as the template for map fitting. Regions corresponding to LRRK2 oligomers and microtubules are colored in aquamarine and light gray, respectively. Higher-resolution, focused maps including both LRRK2 (**C**) and microtubules (**F**) were fitted in (**D**) to generate the composite map (**G**), viewed perpendicular to (left) and along (right) the microtubule axis. (**H–I**) Overview of the 13-pf microtubule map used for the composite map shown in (**F**). Fitting of a published 13-pf microtubule model (**H**) and an α–β tubulin complex model (**I**) into the map is shown in close-up views. (**J**) Overview of the active-like LRRK2 observed from the in situ study on microtubules viewed perpendicular to (left) and along (right) the microtubule axis similar to (**G**). The domain color codes match all previous figures. Scale bar: 5 nm for all panels.

The online version of this article includes the following figure supplement(s) for figure 1:

**Figure supplement 1.** Geometry-based extraction of LRRK2–microtubule subtomograms.

**Figure supplement 2.** Data-processing scheme for the LRRK2$^{I2020T}$ + MLi-2 + microtubule sample.

**Figure supplement 3.** Data-processing scheme for the focused-refined maps from the LRRK2 + MLi-2 + microtubule sample.

flexible (*Watanabe et al., 2020*). Consistent with this, the kinase domain of MLi-2-bound LRRK2RCKW in the in vitro reconstituted filaments adopts the closed conformation (*Snead et al., 2022*). In addition to the COR:COR dimerization interface, the microtubule-associated filaments are formed by a WD40:WD40 interaction between adjacent LRRK2 dimers. While in situ LRRK2 and in vitro LRRK2$^{RCKW}$ share almost identical filament geometry on microtubules, this assembly is incompatible with the autoinhibited form of LRRK2: direct docking of the autoinhibited-LRRK2 dimer model into the microtubule–LRRK2$^{RCKW}$ map results in clashes involving the N-terminal domains. It is not known whether LRRK2 binds to microtubules only in its active state, or whether the autoinhibited state can bind as well. Given the importance of type-I LRRK2 inhibitors in the current PD-related clinical trials, as well as the different effects on LRRK2–microtubule association induced by type-I and type-II LRRK2 inhibitors, it is important to determine whether autoinhibited LRRK2 can bind to microtubules and, if it does, to understand the association structurally. In this study, we present cryo-ET structures of full-length LRRK2 bound to microtubules in the inactive, autoinhibited conformation. Additionally, we explored the effect of PD-linked mutations and type-I and type-II inhibitors in LRRK2 filament formation and structure on microtubules.

## Results
### Architecture of microtubule-bound filaments of inactive LRRK2

To determine whether the autoinhibited form of LRRK2 can bind to and oligomerize on microtubules, we used cryo-electron tomography (cryo-ET) to image purified full-length LRRK2 (LRRK2$^{WT}$) and one of its hyperactive PD-linked mutants (LRRK2$^{I2020T}$) reconstituted with microtubules, either in the absence or presence of type-I (MLi-2) or type-II (GZD-824) LRRK2 inhibitors. We found that both WT and I2020T full-length LRRK2-decorated microtubules in the presence of either inhibitor (*Figure 1—figure supplement 1A, C*). As was the case with the truncated version of LRRK2 (LRRK2$^{RCKW}$), co-polymerization of full-length LRRK2$^{I2020T}$ and microtubules also yielded LRRK2-decorated microtubules (*Snead et al., 2022*) (see methods, *Figure 1—figure supplement 1A*).

To determine the structure of the oligomers formed by full-length inactive LRRK2 on microtubules, we collected cryo-ET tilt series on four samples: (1) LRRK2$^{WT}$ with GZD-824 on preformed microtubules; (2) LRRK2$^{I2020T}$ with MLi-2 on pre-formed microtubules; (3) LRRK2$^{I2020T}$ with MLi-2 co-polymerized with microtubules; and (4) LRRK2$^{WT}$ with MLi-2 on pre-formed microtubules. Reconstructed tomograms from all samples (*Figure 1—figure supplement 1A, C*) showed that all combinations resulted in a similar overall architecture for the microtubule-bound LRRK2, and MLi-2 does not lead to an active conformation with FL-LRRK2 as observed from in-solution LRRK2 structures (*Sanz Murillo et al., 2023*). To corroborate the conformations and interactions of these filaments, we performed subtomogram analysis and obtained subnanometer-resolution structures for all four conditions (*Figure 1—figure supplements 1B and 2*, *Figure 2—figure supplements 1–3*, *Supplementary file 1* and Methods). From dataset (2) (LRRK2$^{I2020T}$ with MLi-2 on pre-formed microtubules), we performed focused refinement from the map including both the microtubule and LRRK2 and obtained a 7.8-Å map of LRRK2 only, which showed the highest LRRK2 resolution after subtomogram analysis (*Figure 1—figure supplement 2*, *Figure 1B*). Microtubule-focused 3D classification showed the existence of microtubules with different protofilament numbers in the sample (*Figure 1—figure supplement 3A*), from which we obtained a 5.9-Å 13-protofilament microtubule map (*Figure 1F, H–I*, *Figure 1—figure supplement 3A–C*), that we used to build a composite map (*Figure 1G*). Both of the focused maps were docked into the microtubule–LRRK2 map to show the entire assembly (*Figure 1C–G*).

The cryo-ET maps we obtained showed distinct features compared to the in situ microtubule-bound LRRK2 map (*Figure 1J*, *Figure 1—figure supplement 2*; *Watanabe et al., 2020*). Thus, we set out to build a molecular model of microtubule-bound full-length LRRK2 filaments using the aforementioned maps (*Figure 1B*). Our model of full-length LRRK2 on microtubules is similar to the published cryo-EM models of in-solution, autoinhibited full-length LRRK2 (*Myasnikov et al., 2021*; *Zhu et al., 2024*; *Sanz Murillo et al., 2023*), with the LRR domain blocking the kinase's active site (*Figure 1B*). As was the case with the structures of in-solution LRRK2,

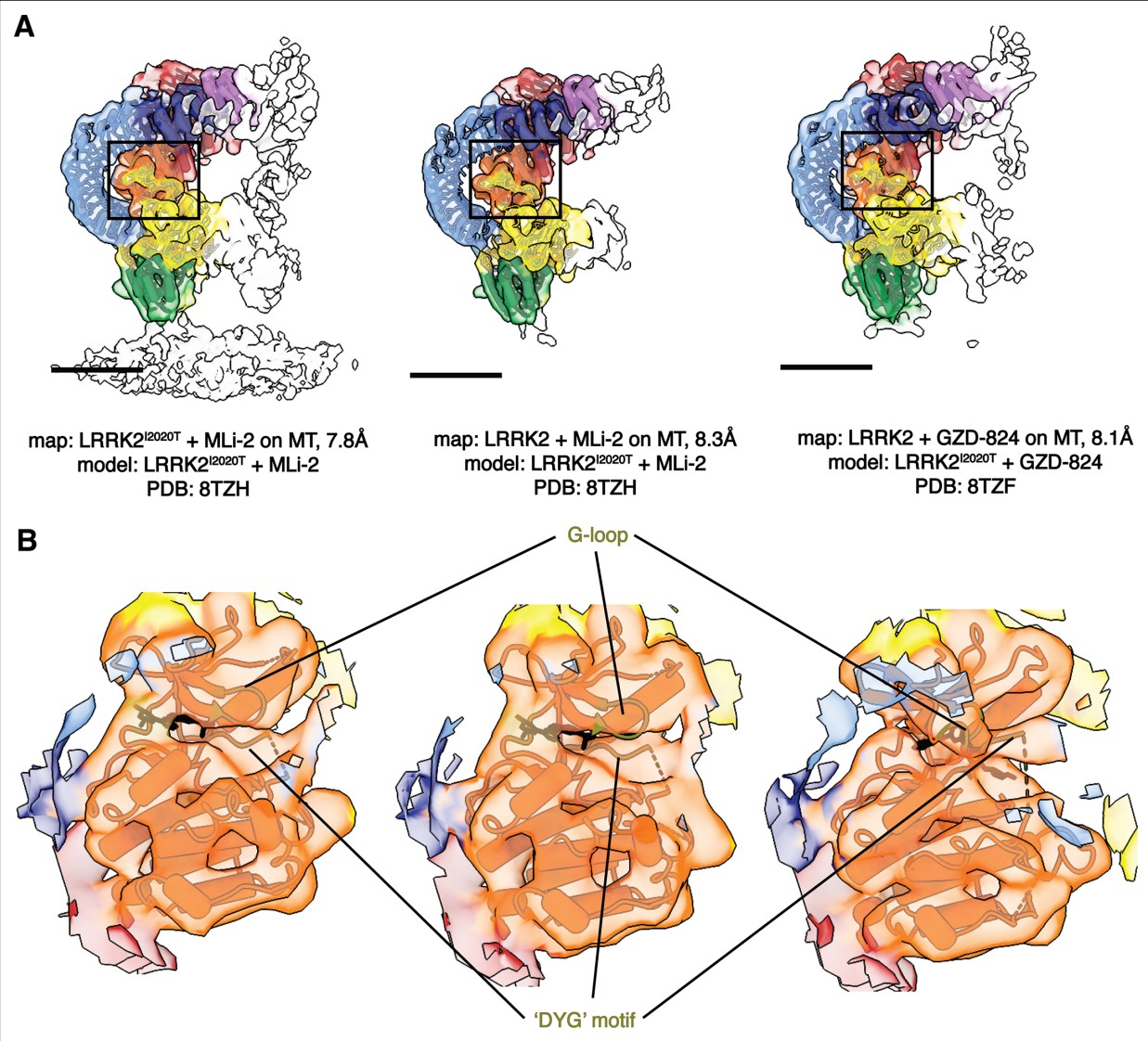

**Figure 2.** Comparison of the LRRK2 kinase domain conformations in the in-solution and microtubule-bound structures. (**A**) Models of in-solution LRRK2[I2020T] bound to kinase inhibitors were fitted into our maps of LRRK2 bound to microtubules. The color scheme is the same one introduced in *Figure 1*. The published LRRK2 models that were fitted into each map contain the same kinase inhibitor used in the sample giving rise to the map, as noted below each image. (**B**) Close-up view focusing on the fitting of the kinase from the published LRRK2–inhibitor complexes into each of our LRRK2 maps. The motifs that are key elements in the kinase and show well-fitting between the maps and the models, e.g. the 'G-loop' and the 'DYG' motif, are highlighted and colored in olive green. The inhibitors are shown in black.

The online version of this article includes the following figure supplement(s) for figure 2:

**Figure supplement 1.** Data-processing scheme for the LRRK2[I2020T] + MLi-2 + microtubule sample with LRRK2[I2020T] and microtubule co-polymerization.

**Figure supplement 2.** Data-processing scheme for the LRRK2 + MLi-2 + microtubule sample.

**Figure supplement 3.** Data-processing scheme for the LRRK2 + GZD-824 + microtubule sample.

the ARM domain is flexible in our map (*Sanz Murillo et al., 2023*). While the resolution of our maps is not sufficient to resolve side chains or determine whether inhibitors are bound, we compared the region of the map corresponding to the kinase domain to the recent structures of in-solution LRRK2 bound to either MLi-2 or GZD-824 (*Zhu et al., 2024*; *Sanz Murillo et al., 2023*). In agreement with those structures and the proposal that the N-terminal repeats constrain conformational changes in LRRK2, the kinase in our full-length LRRK2 adopted an inactive, open conformation in the presence of the type-I inhibitor MLi-2. In the published high-resolution structures of in-solution LRRK2, the 'DYG' motif and G-loop show different

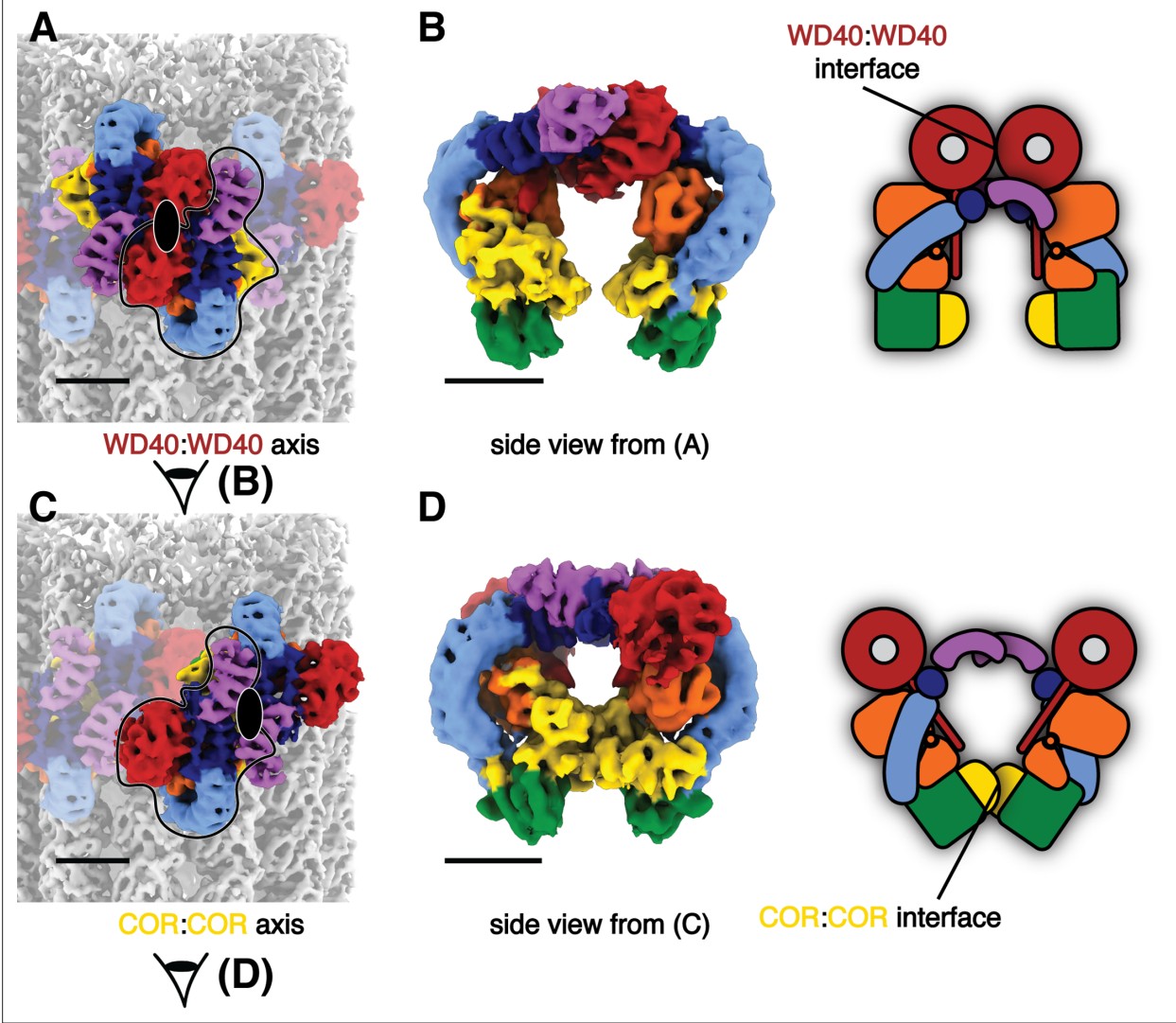

**Figure 3.** The two pseudo-twofold interfaces mediating filament formation of the autoinhibited LRRK2$^{I2020T}$ on microtubules. (**A**) Two LRRK2 copies involved in the WD40:WD40 pseudo-twofold interface are highlighted in color with the rest of the map dimmed. The axis is indicated by a black ellipsoid. (**B**) Side view of the LRRK2 WD40:WD40 dimer from the composite map and its cartoon representation. The view direction is shown as eye symbols in (**A**). WD40:WD40 interaction surface is highlighted in the cartoon representation. (**C**) Similar to (**A**), showing the COR:COR LRRK2 dimer and axis. The same single copy of LRRK2 is highlighted in black contours for reference. (**D**) similar to (**B**), showing the side view and cartoon representation of LRRK2 COR:COR dimer from the composite map.

conformations in the presence of type-I or type-II inhibitors. We observed similar conformations (*Figure 2*), suggesting that full-length LRRK2, whether LRRK2$^{WT}$ or LRRK2$^{I2020T}$, bound to microtubules is in its autoinhibited state in our reconstituted system. Similar to the structure of LRRK2$^{RCKW}$ on microtubules (*Snead et al., 2022*), our model showed two pseudo-twofold symmetry axes, one centered at the COR:COR interface between two LRRK2s in a dimer, and a WD40:WD40 interface between neighboring dimers (*Figures 1G and 3*, *Figure 4—figure supplement 1A*).

## LRRK2's N-terminal repeats interact with the WD40 domain in the microtubule-bound oligomers

To investigate the architecture of the inactive LRRK2 oligomers on microtubules, we fitted the model of the in-solution LRRK2 dimer (*Myasnikov et al., 2021*) into our map and focused on the interface between two in-solution LRRK2 dimers (*Figures 1G and 3*). The WD40 domain of one LRRK2 monomer (copy I) clashes with the ARM domain of a monomer in the neighboring dimer (copy II)

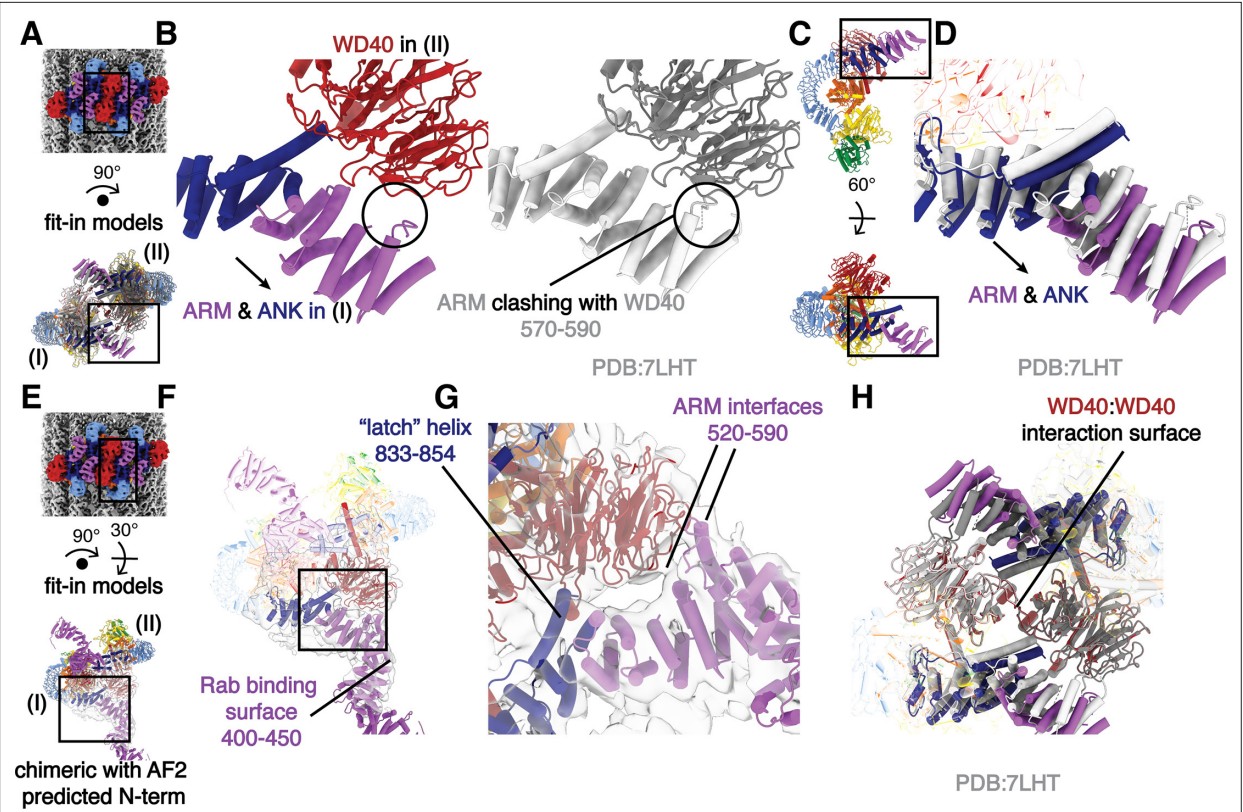

**Figure 4.** The filaments of autoinhibited LRRK2 are stabilized by an interaction between WD40 and ARM/ANK domains. (**A**) Composite map (top) and rotated LRRK2 WD40:WD40 dimer model (bottom). The black squares indicate the location of the close-up view shown in (**B**). As labeled, the left and right copies of LRRK2 are named I and II, respectively. (**B**) Side-to-side comparison of the autoinhibited LRRK2 WD40:WD40 dimer model on microtubules (color, left) and the aligned in-solution LRRK2 model (gray, right). Both models share the same orientation as the model in (**A**). The clashes that result from docking the in-solution LRRK2 models into our map are highlighted by a black circle and are resolved in the autoinhibited LRRK2 WD40:WD40 dimer model on microtubules by a shift in the N-terminal repeats shown with an arrow. (**C**) The location of the close-up view shown in (**D**) is indicated by black squares on the LRRK2 model before (up) and after rotation (down). The original viewing angle before rotation is the same as in *Figure 1B*. (**D**) Superimposition of the LRRK2 structure in solution (light gray) with the autoinhibited LRRK2 on microtubules (colored). The shift of the ARM/ANK repeats is shown with an arrow. Both models share the same viewing angle as the rotated model in (**C**) at the bottom. (**E**) Composite map (top) and rotated full-length LRRK2 WD40:WD40 dimer model (bottom). The black squares indicate the location of the close-up view shown in (**F**). The model combined the model of autoinhibited LRRK2 we built into our map and the Alpha-Fold predicted N-terminus (see Methods). Similar to (**A**), the left and right copies of LRRK2 are named as copy I and copy II, respectively. (**F**) The chimeric full-length LRRK2 model was fitted into the focused refinement map (light gray). The map was refined by creating a focused mask on the WD40 domain of LRRK2 copy I and the ARM/ANK domains of LRRK2 copy II involved in the protein–protein interface (see Methods). The Rab-binding site at the N-terminus of LRRK2 is highlighted. (**G**) Close-up view of the binding surfaces between ARM/ANK domains and the WD40 domain in the autoinhibited-LRRK2 dimer model on microtubules. The 'latch' helix and the WD40-binding interfaces in the ARM domain are highlighted based on the focused refinement map. The black square in (**F**) highlights the focused region shown here. (**H**) Superimposition of the LRRK2 in-solution model (light and dark gray) and the autoinhibited-LRRK2 WD40:WD40 dimer model on microtubules (colored). The two models were aligned by their WD40 domains. The view is focused on the ARM/ANK and WD40 domains of both LRRK2 copies, and the rest of the models are shown as semi-transparent. Scale bar: 5 nm for all panels.

The online version of this article includes the following figure supplement(s) for figure 4:

**Figure supplement 1.** Surface electrostatic potential at the WD40:ARM/ANK interaction interfaces in the LRRK2 WD40 dimer observed in the LRRK2–microtubule complex.

after model fitting (*Figure 4A, B*). We resolved these clashes by shifting the N-terminal ARM/ANK domains of each LRRK2 away from its own WD40 domain (*Figure 4*, *Figure 1—figure supplement 3*). This suggests that a novel WD40:ARM interaction is involved in stabilizing the oligomerization of autoinhibited full-length LRRK2 on microtubules.

In order to better characterize this new interface, we performed focused refinement and 3D classification around the WD40:ARM/ANK region. This revealed more density corresponding to the N-terminal ARM repeats of LRRK2, including several interactions between the ARM/

ANK domain and the WD40 domain; this region of the ARM domain is close to the reported Rab29-binding site (*Zhu et al., 2023*) but does not overlap with it. The fact that none of the interactions bury large surfaces, and that we could only resolve the additional density, which has lower local resolution, after focused classification, suggests that the ARM/ANK domains of autoinhibited LRRK2 are still flexible in the oligomers (*Figure 4E, F*, *Figure 1—figure supplement 3D–F*). The surface charge distribution of both the WD40 and ARM/ANK domains shows that both hydrophobic and charged interactions are likely to be involved in these interactions (*Figure 4—figure supplement 1D–H*). Interestingly, the 'latch' helix in the LRR, which interacts with its own WD40 domain in the full-length LRRK2 structure, has its C-terminal end pointed toward a positively charged patch on the WD40 surface from the other LRRK2 dimer (*Figure 4G*, *Figure 4—figure supplement 1G–H*). The WD40:WD40 interaction in our structure is similar to that of the microtubule-bound LRRK2$^{RCKW}$ filaments (*Figure 4H*).

## Autoinhibited-LRRK2 forms short and sparse oligomers on microtubules

Since we extracted subtomograms from the cryo-ET data, we could map the subtomograms that contributed to the final reconstruction back to the original tomograms and analyze their distributions around microtubules (*Figure 1—figure supplement 1C*, *Figure 5A*). We selected WD40:WD40 LRRK2 dimers (*Figures 1F and 3A, B*) as the species to carry out this analysis (*Figure 5*, *Figure 5—figure supplement 1A, B*) because we used the same species to analyze the WD40:ARM/ANK interface. This revealed that the helical angle between the previously reported microtubule-bound filaments of active-like LRRK2 or LRRK2RCKW and our filaments of autoinhibited LRRK2 is different. The distribution of these angles in our samples peaked at around 90° ($\theta$ angle, *Figure 5E*), leading to ring-like structures around the microtubule, in contrast to the right-handed helices with helical angles around 33° observed with the active-like forms of LRRK2 (*Snead et al., 2022*; *Watanabe et al., 2020*). In addition, our analysis of the distribution of LRRK2 subtomograms showed that the LRRK2 oligomers do not form continuous filaments wrapping around microtubules. Instead, the most frequently observed species were individual dimers and short oligomers consisting of two to six LRRK2 dimers (*Figure 5F*). Since the COR:COR LRRK2 dimer is the dominant species in solution while we chose the WD40:WD40 dimer as the reference to search the LRRK2 subtomograms, it is worth noting that one copy of an LRRK2 monomer may be missed at the edge of the LRRK2 groups, or that only one copy of an LRRK2 monomer from the picked WD40:WD40 dimer on the edge contributed to the refinement. Both cases may happen within a single LRRK2 subtomogram that does not belong to a multi-subtomogram group, thus resulting in individual LRRK2 monomers or part of a COR:COR dimer. The results were consistent across all datasets, regardless of the LRRK2 variant used, the type of inhibitor added, or the protocol used to generate the microtubule-bound filaments (*Figure 5—figure supplement 1*). Notably, this is in contrast to the highly ordered long-range filaments observed in the LRRK2$^{RCKW}$–microtubule data. The only minor difference we observed in the sample containing GZD-824 was that LRRK2 formed longer oligomers more frequently than in the presence of MLi-2. However, none of them covered more than half a turn around the microtubule (*Figure 5—figure supplement 1E*), and this difference does not increase the coverage of LRRK2 decorations on microtubules in our sample, as shown by the LRRK2 coverage analysis (*Figure 5—figure supplement 1F*).

## Discussion

Using an in vitro reconstituted system and cryo-ET approaches followed by subtomogram analysis, we have shown that the autoinhibited form of LRRK2 is capable of forming oligomers on microtubules. This assembly shares features with the previously reported structures of active-like LRRK2 bound to microtubules, but also shows different helical parameters and entails a new interface involving the N-terminal repeats, which, in contrast to the filaments formed by active-like LRRK2, are ordered in this structure. Our previously published active-like LRRK2 structures of in situ (LRRK2) and in vitro reconstituted (LRRK2$^{RCKW}$) microtubule-bound filaments formed right-handed helices with the same interfaces and helical parameters. In the in situ structure, the N-terminal repeats were flexible and not visible in

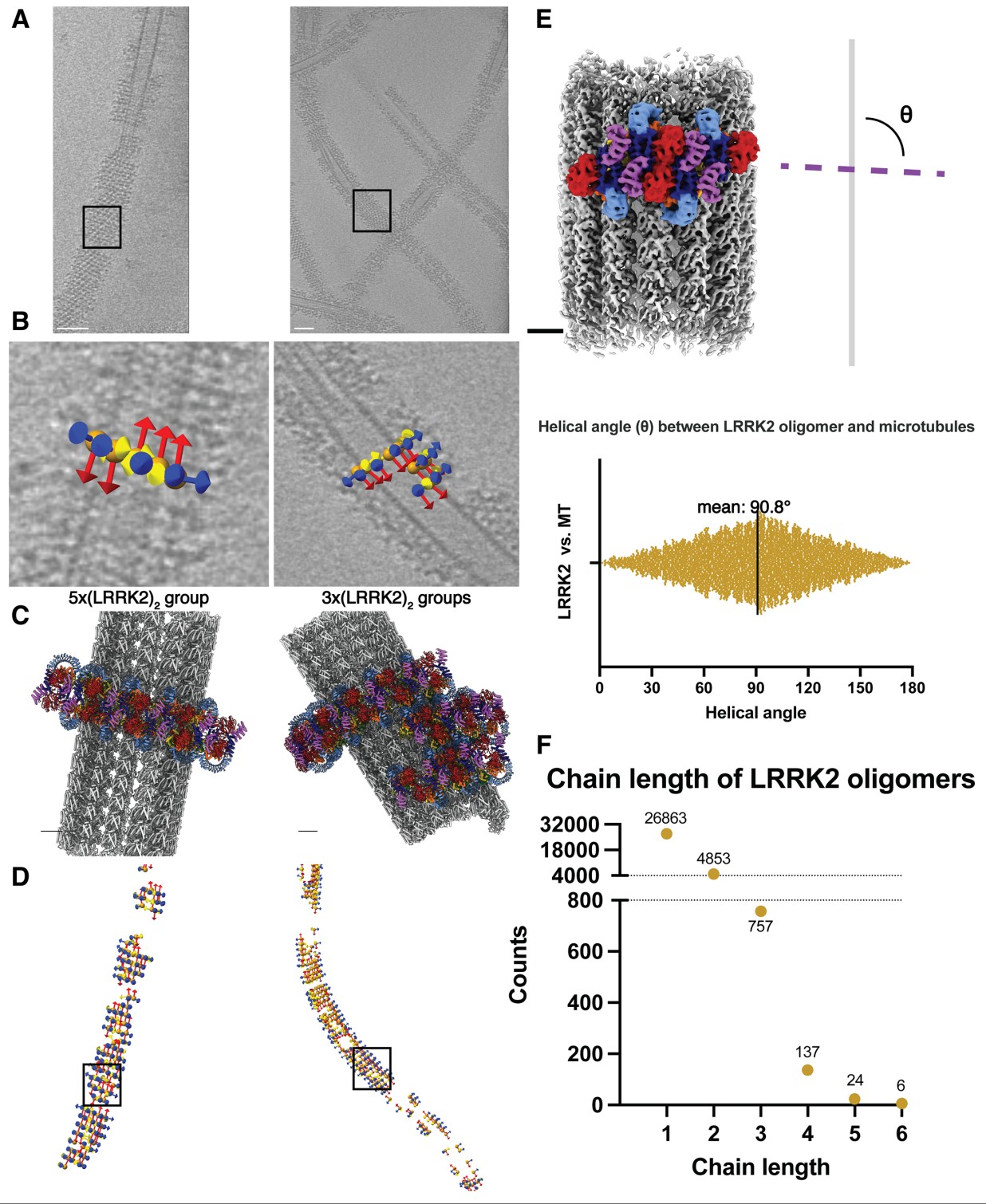

**Figure 5.** Autoinhibited-LRRK2 forms short, sparse oligomers perpendicular to the microtubule axis. (**A**) Tomographic slices 4.6 nm in thickness showing LRRK2-decorated microtubules. (**B**) Close-up views showing subtomogram coordinates (orange spheres) and their refined *x–y–z* (yellow–red–blue) orientations after subtomogram analysis. The picked subtomograms are mapped back to the original tomogram, and the LRRK2 dimer groups containing different copy numbers are shown (left: 5-copy group, right: several 3-copy groups clustering together). Black squares in (**A**) and (**D**) highlight the locations of corresponding subtomogram groups. (**C**) Composite model showing LRRK2 oligomer groups binding to microtubules, with orientations and LRRK2 copy numbers corresponding to (**B**). (**D**) Location and orientation of picked subtomograms contributing to the final LRRK2 reconstruction, color-coded in the same way as in (**B**). The subtomogram picking models are aligned to their corresponding microtubules in (**A**). (**E**) Representation of

*Figure 5 continued on next page*

*Figure 5 continued*

the definition of the helical angle (*θ* angle), showing the angle between (LRRK2)₂ oligomers and the axis of the microtubule (above), as well as the plot showing the distribution of $\theta$ angles observed from this dataset (below). (**F**) Picked LRRK2 subtomograms are grouped based on their relative positions and distances, and the length of (LRRK2)₂ oligomers grouped together is plotted as a frequency plot. The longest LRRK2 dimer chain observed in this dataset contains six LRRK2 dimers. Scale bars: (**A**) 50 nm, (**B–E**) 5 nm.

The online version of this article includes the following figure supplement(s) for figure 5:

**Figure supplement 1.** Geometrical analysis of LRRK2 subtomograms for the different LRRK2–microtubule datasets.

the subtomogram averages. The kinase domain adopted a closed, active-like conformation in both of those structures (*Snead et al., 2022*; *Watanabe et al., 2020*). In contrast, our in vitro reconstituted microtubule-bound filaments of autoinhibited LRRK2 revealed a stabilized N-terminal half (ARM/ANK/ LRR domains) covering the kinase's active site. The kinase domain in our structure was in the open, inactive conformation, as expected from the autoinhibited state of LRRK2 (*Myasnikov et al., 2021*; *Zhu et al., 2024*; *Sanz Murillo et al., 2023*; *Zhu et al., 2023*). Our data showed that LRRK2 is capable of binding to microtubules in its autoinhibited state, although it appears to do so without forming the long, continuous filaments observed with its active form (*Figure 6*). Notably, we previously demonstrated that active-like LRRK2, when bound to a type-I inhibitor, can form roadblocks that impair vesicular transport. Since autoinhibited LRRK2 assembles into shorter, less stable oligomers on microtubules, we anticipate it will exert reduced road-blocking effects in cells, regardless of the inhibitor bound.

Understanding how type-I and type-II inhibitors' binding to LRRK2 affects its mechanism is vital to the design of inhibitor-based PD drug development strategies. Our findings revealed that different LRRK2 kinase inhibitors bind to autoinhibited LRRK2 similarly either in solution or on microtubules. Furthermore, the observation of autoinhibited-LRRK2 forming short, less stable oligomers on microtubules opens new possibilities to inhibit LRRK2 activity in PD patients. A type-I inhibitor specifically targeting autoinhibited LRRK2 may alleviate the effect of LRRK2 roadblocks on microtubules. Alternatively, a promising strategy of LRRK2 inhibitor design can focus on the stabilization of allosteric N-terminus blocking on the kinase domain, which favors the formation of autoinhibited-LRRK2 oligomers on microtubules and causes fewer side effects.

The new WD40:ARM/ANK interface we identified near the WD40:WD40 pseudo-twofold axis showcased again the capability of LRRK2 to form different types of oligomers (*Myasnikov et al., 2021*; *Zhu et al., 2023*; *Watanabe et al., 2020*), and further addressed the importance of the scaffolding function of the LRRK2 WD40 domain. The biological role of this interface and whether PD-related LRRK2 mutations affect the WD40:ARM/ANK interaction remains to be determined. Genetic analysis of PD patients revealed two missense mutations, K544E (*Xiromerisiou et al., 2007*) and N551K (*Di Fonzo et al., 2006*; *Hui et al., 2018*), located on the same α-helix near one of the interfaces on the N-terminal repeats (*Figure 4—figure supplement 1I*). N551K was reported to confer protection to both PD and Crohn's disease (*Hui et al., 2018*; *Kalogeropulou et al., 2022*), and an overexpression study showed a mild but statistically significant negative impact of N551K on MLi-2-induced association between LRRK2 and microtubules (*Kalogeropulou et al., 2022*). Our data suggests a need to understand if and how LRRK2 interacts with microtubules under physiological conditions, and further, whether LRRK2 is able to alternate its oligomerization states on microtubules in the cell, a possibility raised by this and our previous studies (*Snead et al., 2022*; *Watanabe et al., 2020*) which show that LRRK2 can interact with microtubules in more than one conformation. LRRK2 plays multiple roles in the endolysosomal system, such as phosphorylation of Rab-GTPases and homeostasis maintenance when lysosomes are stressed (*Steger et al., 2016*; *Bonet-Ponce et al., 2020*; *Eguchi et al., 2018*). LRRK2 has also been shown to bind and reshape a negatively charged tubular membrane template in vitro (*Wang et al., 2023*). Our findings further demonstrate the ability of LRRK2 to oligomerize on microtubules in both active-like and inactive conformations. Given the geometric and electrostatic similarities between microtubules and tubular membrane structures, we speculate that this may also occur on membranes where LRRK2 performs its physiological functions.

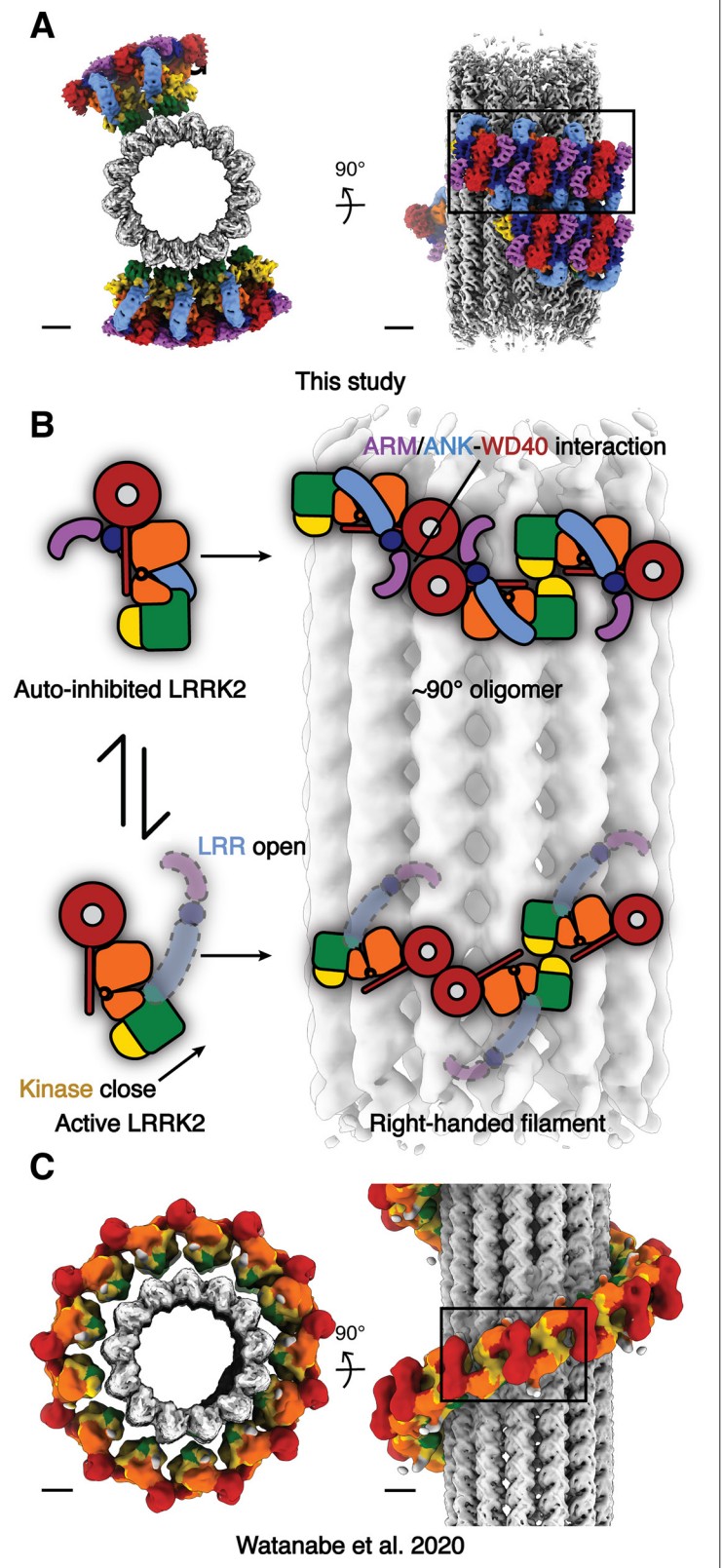

**Figure 6.** Summary of the comparison between the autoinhibited-LRRK2 state and the active-like LRRK2 state on microtubules. (**A**) Autoinhibited-LRRK2 filaments observed from this study. (**B**) The cartoon representations describing the differences of the domain architectures between the autoinhibited and the active-like LRRK2, as well as the corresponding different assemblies observed on microtubules. (**C**) The active-like LRRK2 right-handed

*Figure 6 continued on next page*

*Figure 6 continued*

filaments observed from the in situ study. The map is derived from the published maps with color codes matching all previous figures. Scale bars: 5 nm.

# Materials and methods

## Key resources table

| Reagent type (species) or resource | Designation | Source or reference | Identifiers | Additional information |
|---|---|---|---|---|
| Other | Models and maps of LRRK2$^{I2020T}$ with MLi-2 on pre-formed microtubules | PDB data bank | PDB: 9CHO, EMD-45591, EMD-45592, EMD-45593 | Cryo-ET dataset |
| Other | Maps of LRRK2I2020T with MLi-2 co-polymerized with microtubules | PDB data bank | EMD-45594 | Cryo-ET dataset |
| Other | Maps of LRRK2WT with MLi-2 on pre-formed microtubules | PDB data bank | EMD-45595 | Cryo-ET dataset |
| Other | Maps of LRRK2WT with GZD-824 on pre-formed microtubules | PDB data bank | EMD-45596 | Cryo-ET dataset |
| Software, algorithm | serialEM | University of Colorado, Boulder | http://bio3d.colorado.edu/SerialEM/, RRID:SCR_017293 | Cryo-ET data collection software |
| Software, algorithm | Warp | Max Planck Institute | http://www.warpem.com/warp/, RRID:SCR_018071 | Cryo-ET data-processing software |
| Software, algorithm | IMOD | University of Colorado, Boulder | https://bio3d.colorado.edu/imod/, RRID:SCR_003297 | Cryo-ET data-processing software |
| Software, algorithm | Dynamo in Matlab | University of Basel | https://www.dynamoem.org//w/index.php?title=Main_Page, RRID:SCR_025592 | Cryo-ET data-processing software |
| Software, algorithm | Commands used in Dynamo | Self | https://doi.org/10.5281/zenodo.13314157 | Commands used in Dynamo |
| Software, algorithm | Relion | MRC Laboratory of Molecular Biology | http://www2.mrc-lmb.cam.ac.uk/relion, RRID:SCR_016274 | Cryo-ET data-processing software |
| Software, algorithm | ChimeraX | UCSF | https://www.cgl.ucsf.edu/chimerax/, RRID:SCR_015872 | Cryo-ET data-processing and visualization software |
| Software, algorithm | ISOLDE | Altos Labs | https://tristanic.github.io/isolde/, RRID:SCR_025577 | Cryo-ET model building software |
| Other | LRRK2 cloning, plasmid construction, and mutagenesis | https://www.protocols.io/ | https://www.protocols.io/view/lrrk2-cloning-plasmid-construction-and-mutagenesis-kxygx35ddg8j/v1 | Protocol |
| Other | Insect Cell Protocol for LRRK1 and LRRK2 Expression | https://www.protocols.io/ | | Protocol |
| Other | LRRK2 RCKW Protein Purification | https://www.protocols.io/ | https://www.protocols.io/view/lrrk2-rckw-protein-purification-81wgb6693lpk/v1 | Protocol |
| Other | Reconstituting LRRK2RCKW on Microtubules for cryo-EM studies | https://www.protocols.io/ | https://www.protocols.io/view/reconstituting-lrrk2rckw-on-microtubules-for-cryo-3byl4kjb8vo5/v1 | Protocol |
| Other | Quick guide to use paceTOMO for cryo-ET data collection from Titan Krios | https://www.protocols.io/ | https://www.protocols.io/view/quick-guide-to-use-pacetomo-for-cryo-et-data-colle-6qpvr3442vmk/v1 | Protocol |
| Other | Tomogram reconstruction and subtomogram analysis of LRRK2 on microtubules | https://www.protocols.io/ | https://www.protocols.io/view/tomogram-reconstruction-and-sub-tomogram-averaging-n92ld89zxv5b/v2 | Protocol |

## Purification of protein factors

Full-length LRRK2, both wild-type and I2020T mutant, was purified from Sf9 cells as previously described (*Deniston et al., 2020*). The codon-optimized full-length LRRK2 sequence was inserted

into the same vector used previously via Gibson assembly and LRRK2 mutant as cloned using the Q5 Site-Directed Mutagenesis Kit (NEB) following the manufacturer's instructions. Our general protocols to prepare plasmids and purify LRRK2 and their variants from insect cells are available at https://www.protocols.io/view/lrrk2-cloning-plasmid-construction-and-mutagenesis-kxygx35ddg8j/v1, https://www.protocols.io/view/insect-cell-protocol-for-lrrk1-and-lrrk2-expressio-rm7vzyyrrlx1/v1, and https://www.protocols.io/view/lrrk2-rckw-protein-purification-81wgb6693lpk/v1. Cells were first washed with phosphate-buffered saline, then lysed by homogenization in lysis buffer [50 mM Hepes (pH 7.4), 500 mM NaCl, 20 mM imidazole, 0.5 mM TCEP, 5% glycerol, 5 mM $MgCl_2$, and 20 µM GDP]. After centrifugation, the supernatant was first purified by a Ni-NTA (Qiagen) column since the LRRK2 construct includes cleavable His6-ZZ tags. The sample was then eluted in lysis buffer containing 300 mM imidazole. Next, the salt concentration in the eluate was diluted to 250 mM NaCl with dilution buffer [50 mM Hepes (pH 7.4), 0.5 mM TCEP, 5% glycerol, 5 mM $MgCl_2$, and 20 µM GDP] and loaded onto an SP Sepharose column. The elution was performed with a salt gradient from 250 mM to 2.5 M NaCl. The eluate was then treated overnight with Tobacco Etch Virus (TEV) protease to cleave the $His_6$-ZZ tag. On the next day, an SP His-trap column is used to further purify the target proteins. After that, the samples were concentrated and subjected to an S200 gel filtration column in an ÄKTAxpress system. The buffer for wild-type LRRK2 purification contains 20 mM Hepes (pH 7.4), 700 mM NaCl, 0.5 mM TCEP, 5% glycerol, 2.5 mM $MgCl_2$, and 20 µM GDP, while $LRRK2^{I2020T}$ purification used 200 mM NaCl. The eluted samples are concentrated by centrifuge when necessary and final yields of the pure proteins are calculated from ultraviolet absorbance. The final concentrations of wild-type LRRK2 and $LRRK2^{I2020T}$ were 9.8 and 3.14 mg/ml, respectively.

## Assembly of the LRRK2 on microtubules

The assembly of LRRK2 on microtubules was performed similarly as previously described with some critical optimizations (*Snead et al., 2022*). The general protocol is also available at https://www.protocols.io/view/reconstituting-lrrk2rckw-on-microtubules-for-cryo-3byl4kjb8vo5/v1. The final molar ratio between LRRK2 and tubulin was 4:1 (5 µM LRRK2 and 1.25 µM tubulin in the final sample). Unpolymerized mouse brain tubulin was first polymerized in polymerization buffer [80 mM PIPES, 2.5 mM $MgCl_2$, 1 mM EGTA, 1 mM DTT, 1 mM GTP, 10 µM taxol, and 5% glycerol, pH 7.0] for 45 min. Meanwhile, purified LRRK2 (wild-type or I2020T mutant) was co-incubated with 10 µM of either MLi-2 or GZD-824 for 5 min in polymerization buffer containing 150 mM salt (salt came from the LRRK2 stock buffer). After that, LRRK2 was mixed with previously polymerized tubulin to co-incubate for 15 min. The salt concentration in this sample is 90–100 mM, which ensures efficient tubulin polymerization as well as less LRRK2 aggregation. In the case of the co-polymerization sample, microtubules and drug-treated LRRK2 were added together with the same ratio and final salt concentration for 1 hr. The two protocols resulted in similar assembly of LRRK2 on the microtubules and yielded similar maps from subtomogram analysis.

To prepare samples for cryo-ET data acquisition, the assembled LRRK2–microtubule complexes are diluted with LRRK2 reaction buffer [20 mM HEPES, 80 mM NaCl, 0.5 mM TCEP, 2.5 mM $MgCl_2$, 10 µM Taxol, pH 7.4] in a 1:1 ratio to lower down the glycerol concentration.

## Electron microscopy

To prepare grids for cryo-ET data acquisition, 300 mesh Lacey Carbon copper grids (Electron Microscopy Sciences) were used. Before sample deposition, the grids were plasma-cleaned for 60 s using a Pelco plasma cleaner equipped with air at 25 W power. 4 µl of the diluted sample was then immediately applied to the grid and blotted by a Vitrobot with a force of 3 for 4 s, plunge-frozen in liquid ethane, and stored in liquid nitrogen.

For all datasets collected on the LRRK2–microtubule complex, a Titan Krios-3 TEM (Thermo Fisher) operating at 300 kV with K3 direct electron detector (Gatan) at a nominal magnification of ×42,000 (pixel size 2.161 Å, counting mode). A 70 µm objective aperture was inserted. A zero-loss imaging filter with a 15-eV-wide slit was used throughout the data collection. The defocus range was –3 to –5 µm. paceTOMO (*Eisenstein et al., 2023*) in serialEM (*Mastronarde, 2005*) (http://bio3d.colorado.edu/SerialEM/, RRID:SCR_017293) was used for collecting tilt series automatically. The selection of targets was performed manually, maximizing the number of decorated microtubules while avoiding overlapping microtubules, carbon edges, and empty areas. For each tilt series, a dose-symmetric

scheme was applied with a tilt range of –54° to +54° with an increment of 3°, resulting in 37 tilt images being collected with 8 frames in each tilt. The total dose was 120 e⁻/Å² and the dose rate was set at 5 e⁻/Å²/s. The protocol describing detailed steps in data collection is available at dx.doi.org/10.17504/protocols.io.6qpvr3442vmk/v1.

## Tilt series alignment and tomogram reconstruction

Warp (*Tegunov and Cramer, 2019*) (http://www.warpem.com/warp/, RRID:SCR_018071) was used for the pre-processing of tomograms before subtomogram analysis. Specifically, motion correction, CTF estimation, and gain correction were performed in Warp and then all tilt series were aligned in eTomo in IMOD (*Mastronarde and Held, 2017*) (v4.11.25, https://bio3d.colorado.edu/imod/, RRID:SCR_003297) using patch tracking. To ensure better alignment on true features rather than the carbon substrate and empty space, tilt series were manually examined to preserve patches only in regions containing microtubules. The residual error mean (in nm) calculated from eTomo was used to assess the alignment quality for different tilt series. Tomograms with <2 nm residual error mean are eligible for future analysis. Next, the eTomo alignment data was fed back to Warp for tomogram reconstruction using weighted back-projection at 10 Å/pxl. Tomogram visualization and microtubule tracing were performed on deconvoluted tomograms generated from Warp with default settings.

## Microtubule tracing, LRRK2 particle picking, and ab initio model building in Dynamo

IMOD was used to visualize all reconstructed tomograms and manually trace the backbone of microtubules. Only microtubules with LRRK2 decoration and without extensive overlapping were selected for further analysis. The coordinates of selected microtubules were analyzed in Dynamo in MATLAB (v11509, https://www.dynamo-em.org/w/index.php?title=Main_Page, RRID:SCR_025592) (*Castaño-Díez et al., 2017*) and re-sampled equidistantly for particle extraction from the reconstructed tomogram. The azimuth angle around the microtubule of the picked subtomograms was randomized to minimize the effect of the missing wedge as performed in our previous studies (*Watanabe et al., 2020*; *Croxford et al., 2021*). Next, the picked subtomograms were aligned with a featureless tubular reference generated from the class average of microtubule particle stack without alignment, with local parameters limiting the first two Euler angles and movements along the microtubule. Duplicate subtomograms were then removed based on distance, and coordinates were smoothed again to be equidistantly distributed for LRRK2 particle picking in the next step. The box size of subtomograms was 48 nm in this step.

To pick LRRK2 subtomograms in Dynamo, coordinates of microtubules were used as reference points to pick subtomograms in rings orthogonal to the path, pointing outwards. LRRK2 subtomograms were over-picked, with a radius of 23 nm to the center, particle separation of 7 nm between ring layers, and 18 subunits per ring (*Figure 1—figure supplement 1A*). With this set of parameters, both LRRK2 and about half of the microtubule filaments are included with a particle box size of 36 nm. Next, subtomograms from a microtubule with relatively full LRRK2 decoration were manually chosen to build the initial model (*Figure 1—figure supplement 1*), with subtomograms aligned in Dynamo for 10 iterations without any initial reference. This low-resolution initial model was used for all subtomogram analysis in this study. All LRRK2 subtomograms were then aligned to the reference for only one iteration in Dynamo, and both duplicate subtomograms as well as subtomograms having lower than 0.32 cross-correlation value against the reference were removed. Meanwhile, the average density from each microtubule was manually examined using IMOD to check the skewing direction of tubulin filaments from *z*-direction. The orientation of traced microtubules was random, so subtomograms with opposite skewing directions were flipped in Dynamo to align microtubules better. Meanwhile, the refined particle stack was centered to the WD40:WD40 interaction surface of LRRK2 oligomer for better alignment. The selected subtomograms were then subjected to four iterations of rough and local refinement in Dynamo, resulting in relatively regular but scarcely distributed oligomers around the microtubule (*Figure 1—figure supplement 1C*). The coordinates were then used for further subtomogram analysis in Relion3 (*Zivanov et al., 2018*) and Relion4 (*Zivanov et al., 2022*) (http://www2.mrc-lmb.cam.ac.uk/relion, RRID:SCR_016274).

## Subtomogram analysis and three-dimensional model reconstruction

For the 3D reconstruction of the LRRK2$^{I2020T}$–MT complex, 65,612 subtomograms were supplied to Relion3 after Warp particle extraction. The subtomograms were binned by 2 at this stage (4.322 Å/pxl). Subtomograms were refined in Relion3 with C1 symmetry first, and the middle row of the LRRK2 oligomer was masked to perform focused refinement. To further improve resolution, coordinates of refined subtomograms were used to extract subtomograms in Relion4, then Tomo-frame alignment and CTF refinement were applied to optimize the tilt alignment for each tomogram. Satisfyingly, the map resolution improved, reaching the Nyquist limit after these refinement steps (*Figure 1—figure supplement 2A*). Next, focused refinement was applied to the middle 2 copies of LRRK2 followed by a 10-class 3D classification without any alignment. To avoid overfitting, the regularization parameter T was set to 1 instead of default 4, and the resolution E-step was set to 20 Å to disable alignment of high-frequency signals. Class averages containing at least one good-looking LRRK2 were all chosen for further alignment, subjected to a C2 refinement. Next, one copy of LRRK2 was masked for further refinement, and the stack was symmetry expanded by twofold so that all LRRK2 copies can be included for alignment. A 3D classification with the same set of parameters was applied, resulting in two good classes containing 60,556 LRRK2 subtomograms. The refined stack was unbinned (2.161 Å/pxl) and refined to 8.0 Å, then further improved to 7.8 Å with another round of Tomo-frame alignment, with motions of each subtomograms estimated as well. All reported resolutions correspond to the gold-standard Fourier Shell Correlation using the 0.143 criterion (*Scheres, 2012*). A similar pipeline was applied to another dataset of LRRK2I2020T with MLi-2 on microtubules, with microtubules and LRRK2 co-polymerized with MLi-2 (83,007 subtomograms). However, since the reconstruction shows stripe-like artifacts after initial refinement in Relion3, possibly coming from missing wedges, initial steps of pseudo-tomogram reconstruction in Relion4 were performed with down-weighting particle signals in a 10° cone in Fourier space along the *z* axis (*Figure 2—figure supplement 1*). The particle stacks were resumed to normal after CTF refinement. 45,155 subtomograms from 3D classification finally give an 8.3 Å map, and when combined with the previous stack, the final map resolution was improved slightly to 7.7 Å. The map from the C2 refinement of co-polymerized LRRK2$^{I2020T}$ + MLi-2 + microtubule dataset, called microtubule–LRRK2 full map, was used to make the composed model in *Figure 1*.

The further optimized pipeline was applied to other datasets – LRRK2 with MLi-2 on microtubules (45,052 subtomograms) and LRRK2 with GZD on microtubules (70,363 subtomograms). First, the Relion3 initial alignment steps were skipped, and dynamo stacks were directly used to extract bin-2 subtomograms in Relion4. After that, subtomograms were refined with a full mask covering microtubules and the middle row of LRRK2 oligomers, then subjected to the same pipeline of Tomo-frame alignment, CTF refinement, focused refinement, and 3D classification (*Figure 2—figure supplements 2 and 3*). In the end, 42,687 subtomograms from the LRRK2 + MLi-2+microtubule dataset resulted in an 8.3 Å map and 53,214 subtomograms from the LRRK2 + GZD-824 + microtubule dataset resulted in an 8.1-Å map.

Subtomograms from the stepwise polymerization dataset were used to generate the microtubule map containing 13 protofilaments for the composed model. Since C2 refinement on LRRK2 oligomers may flip the *y*-orientation of LRRK2 subtomograms, a C1 refinement focusing on only one copy of LRRK2 was performed, then followed by a 3D classification to remove bad LRRK2 subtomograms. Next, coordinates of 41,853 subtomograms were processed in Dynamo and geometrically re-centered to focus on the bound microtubule. The *z*-direction of each particle was rotated as well, from being perpendicular to becoming parallel to the microtubule. This way, only microtubules decorated by at least one good copy of LRRK2 can be picked. Next, microtubule subtomograms were extracted in 10 Å/pxl and box size of 72 nm in Dynamo and briefly aligned against a 13-pf microtubule initial model for 1 iteration. The aligned stack was then subjected to a 1-iteration multi-reference alignment in Dynamo using 11–16 protofilament microtubules as reference models. Finally, 6,487 subtomograms fall into the class of 13-pf microtubule, and this sub-stack was subjected back to Relion4 to extract bin-2 microtubule subtomograms with box size of 176 pixels (*Figure 1—figure supplement 3A*). Using a feature-less tube as initial reference, Relion4 helical refinement (with tube diameter 300 Å, twist –27.7°, rise 9.4 Å, central *z*-length 30%, all parameters are optimal for a 13-protofilament microtubule) gives an 8.644-Å map, which is the Nyquist limit of the data. Unbinned subtomograms were then further refined with the same set of parameters, resulting in a 5.9-Å microtubule map.

The same dataset was used to refine the novel WD40:ARM/ANK interaction surface. After getting the 7.8 Å LRRK2 map, a partial soft mask focusing on the ARM/ANK domain and the WD40 domain it interacts with (from the other copy of LRRK2) was applied to the LRRK2 map, and the subtomograms were re-centered, binned by 2, and re-extracted in Relion4 to perform a focused refinement, which resulted in an 8.9-Å focused map (*Figure 1—figure supplement 3B*). Similar to the previous pipeline, a 3-class 3D classification was performed, resulting in one class showing extended density corresponding to the N-terminal region of LRRK2. Such region remained flexible and was not resolved in the LRRK2 map. 28,128 subtomograms from this class were then selected for focused refinement, and the unbinned stack resulted in an 8.6-Å map of the focused region, suitable for further model building.

UCSF ChimeraX (v1.7, https://www.cgl.ucsf.edu/chimerax/, RRID:SCR_015872) was used for all the volume segmentation, figure and movie generation, and automatic rigid-body docking processes (*Pettersen et al., 2021*).

## Model building

Published models of LRRK2 were used as references to build the LRRK2 model on microtubules (*Myasnikov et al., 2021*; *Sanz Murillo et al., 2023*). ChimeraX was first used to rigidly fit each domain of the in-solution full-length LRRK2 model into the map, then the interactive molecular-dynamics flexible fitting software (v1.7, https://tristanic.github.io/isolde/, RRID:SCR_025577; *Croll, 2018*) was used to examine each connecting loop between domains and resolve the clashing and obvious wrong fitting with torsion and distances restrained. An MLi-2 molecule was placed in the model based on the alignment between our model and our in-solution high-resolution LRRK2$^{I2020T}$–MLi-2 model published recently. Using the map focused on the WD40:ARM/ANK interface, we fitted the alphafold2 (*Jumper et al., 2021*) prediction model of the N-terminal region of LRRK2 that is missing in the previous structures as a starting model. ISOLDE was then used with restraints to resolve clashes at the connection between our LRRK2 model and the alphafold2 model, as well as moving a helix that was obviously outside the density in the initial model.

## Geometric analysis of LRRK2 subtomograms on microtubules

For the stepwise polymerization dataset (dataset (1)), LRRK2 subtomograms included in the final map were used, but the coordinates of each subtomogram were extracted from the previous C2 refinement job in the Relion pipeline because they correspond to a WD40:WD40 dimer of LRRK2. The coordinates were re-formatted to be analyzed in Dynamo with a pixel size of 10 Å/pxl. Similar to the pipeline of microtubule map generation, the particle stack was sub-boxed to center on microtubules and each LRRK2 subtomogram has a corresponding microtubule subtomogram. The microtubule subtomograms were briefly aligned against a tubular reference so that subtomograms track the microtubule direction. Next, the distances and orientations of all LRRK2 subtomogram pairs were calculated and plotted. There are two most frequently observed LRRK2 pairs on microtubules, one relatively parallel to the microtubule and the other relatively perpendicular to it. Since the composed LRRK2–microtubule indicates that LRRK2 oligomers are growing roughly perpendicular to the microtubule, only this type of subtomogram pairs was picked for further analysis. Distances between all selected pairs were first plotted to make sure LRRK2 was not over- or under-picked, then selected pairs containing overlapping subtomograms were grouped into chains, and the number of LRRK2 dimer chains containing a certain number of LRRK2 dimers was counted and plotted. For each selected pair, the helical angle (θ angle) between the LRRK2 pair and the z-direction of its corresponding microtubule particle was calculated in Dynamo and plotted (*Figure 3E, F*, *Figure 5—figure supplement 1*). Note that the y-orientation of LRRK2 subtomograms can be either the same as, or opposite to the microtubule growth direction, so the angles range from 0° to 180°. Next, the number of LRRK2 subtomograms obtained from each microtubule was counted and plotted, and the dynamo microtubule tracing coordinates obtained from the previously described pipeline were used to calculate the length of each picked microtubule. Such information was used to calculate and plot the number of LRRK2 subtomograms obtained along a fixed length (1 nm) of microtubules. All three other datasets were analyzed in a similar way.

The detailed protocol describing all of the data analysis steps from tomogram reconstruction to subtomogram geometric analysis is available at https://www.protocols.io/view/tomogram-reconstruction-and-sub-tomogram-averaging-n92ld89zxv5b/v2.

## Acknowledgements

All electron microscopy data were collected at the UCSD Cryo-Electron Microscopy Facility, which was built and equipped with funds from UCSD and an initial gift from the Agouron Institute. We thank M Matyszewski for advice on sample preparation. Molecular graphics and analyses were performed in part with UCSF ChimeraX, developed by the Resource for Biocomputing, Visualization, and Informatics at the University of California, San Francisco, with support from National Institutes of Health R01-GM129325 and the Office of Cyber Infrastructure and Computational Biology, National Institute of Allergy and Infectious Diseases. We thank the UCSD Physics Computing for computational support. This research was funded by Aligning Science Across Parkinson's (grant number ASAP-000519) (AEL and EV) through the Michael J Fox Foundation for Parkinson's Research (MJFF), as well as from the National Science Foundation grant DBI 1920374 (to EV). SC is a JCC-HHMI fellow funded by Jane Coffin Childs Memorial Fund for Medical Research. TB is an AHA Predoctoral Fellowship Awardee supported by American Heart Association. JH was supported by an EMBO long-term postdoctoral fellowship (ALTF 871-2020). EV is a Howard Hughes Medical Institute Investigator.

## Additional information

### Funding

| Funder | Grant reference number | Author |
| --- | --- | --- |
| Aligning Science Across Parkinson's | ASAP-000519 | Siyu Chen<br>Tamar Basiashvili<br>Joshua Hutchings<br>Marta Sanz Murillo<br>Amalia Villagran Suarez<br>Erica Xiong<br>Jaime Alegrio Louro<br>Andres E Leschziner<br>Elizabeth Villa |
| National Science Foundation | DBI 1920374 | Elizabeth Villa |
| Jane Coffin Childs Memorial Fund for Medical Research | | Siyu Chen |
| American Heart Association | | Tamar Basiashvili |
| European Molecular Biology Organization | ALTF 871-2020 | Joshua Hutchings |
| Howard Hughes Medical Institute | | Elizabeth Villa |

The funders had no role in study design, data collection, and interpretation, or the decision to submit the work for publication.

### Author contributions

Siyu Chen, Conceptualization, Data curation, Software, Formal analysis, Funding acquisition, Validation, Investigation, Visualization, Methodology, Writing – original draft, Project administration, Writing – review and editing; Tamar Basiashvili, Data curation, Funding acquisition, Writing – review and editing; Joshua Hutchings, Software, Formal analysis, Methodology; Marta Sanz Murillo, Amalia Villagran Suarez, Erica Xiong, Jaime Alegrio Louro, Data curation, Writing – review and editing; Andres E Leschziner, Elizabeth Villa, Conceptualization, Resources, Supervision, Funding acquisition, Visualization, Project administration, Writing – review and editing

### Author ORCIDs

Siyu Chen ⓘ https://orcid.org/0000-0003-4565-4772
Tamar Basiashvili ⓘ https://orcid.org/0000-0003-0394-3832
Joshua Hutchings ⓘ https://orcid.org/0000-0001-6841-8583

Marta Sanz Murillo (iD) https://orcid.org/0000-0002-6175-9315
Amalia Villagran Suarez (iD) https://orcid.org/0000-0003-3400-2330
Erica Xiong (iD) https://orcid.org/0009-0000-8023-5720
Jaime Alegrio Louro (iD) https://orcid.org/0000-0002-2800-923X
Andres E Leschziner (iD) https://orcid.org/0000-0002-7732-7023
Elizabeth Villa (iD) https://orcid.org/0000-0003-4677-9809

Reviewer #1 (Public review): https://doi.org/10.7554/eLife.100799.3.sa1
Reviewer #2 (Public review): https://doi.org/10.7554/eLife.100799.3.sa2
Reviewer #3 (Public review): https://doi.org/10.7554/eLife.100799.3.sa3
Author response https://doi.org/10.7554/eLife.100799.3.sa4

## Additional files

### Supplementary files

Supplementary file 1. Cryo-electron tomography (cryo-ET) data collection and model statistics.
MDAR checklist

### Data availability

Command-line based Dynamo functions are used in matlab environment, and the step-by-step instructions were described in the correponding protocols. Commands are saved as codes at https://doi.org/10.5281/zenodo.13314157. Cryo-EM density maps have been deposited in the Electron Microscopy Data Bank under accession numbers EMD-45591 (LRRK2I2020T+MLi-2+microtubules, stepwise polymerization), EMD-45594 (LRRK2I2020T+MLi-2+microtubules, co-polymerization), EMD-45595 (LRRK2+MLi-2+microtubules), EMD-45596 (LRRK2+GZD-824+microtubules), EMD-45592 (microtubule map from LRRK2I2020T+MLi-2+microtubules, stepwise polymerization), and EMD-45593 (focused map on WD40:ARM/ANK interaction surface, obtained from LRRK2I2020T+MLi-2+microtubules, stepwise polymerization). Model coordinates have been deposited in the Protein Data Bank under accession numbers 9CHO (LRRK2 from LRRK2I2020T+MLi-2+microtubules, stepwise polymerization).

The following datasets were generated:

| Author(s) | Year | Dataset title | Dataset URL | Database and Identifier |
|---|---|---|---|---|
| Chen S, Villa E, Leschziner AE | 2024 | Autoinhibited full-length LRRK2(I2020T) on microtubules with MLi-2 | https://www.ebi.ac.uk/emdb/EMD-45591 | Electron Microscopy Data Bank, EMD-45591 |
| Chen S, Leschziner AE, Villa E | 2024 | Full-length autoinhibited LRRK2(I2020T) co-polymerized with microtubules and MLi-2 | https://www.ebi.ac.uk/emdb/EMD-45594 | Electron Microscopy Data Bank, EMD-45594 |
| Chen S, Leschziner AE, Villa E | 2024 | Full-length autoinhibited LRRK2 on microtubules with MLi-2 | https://www.ebi.ac.uk/emdb/EMD-45595 | Electron Microscopy Data Bank, EMD-45595 |
| Chen S, Leschziner AE, Villa E | 2024 | Full-length autoinhibited LRRK2 on microtubules with GZD-824 | https://www.ebi.ac.uk/emdb/EMD-45596 | Electron Microscopy Data Bank (EMDB), EMD-45596 |
| Chen S, Leschziner AE, Villa E | 2024 | 13-pf microtubule from the LRRK2(I2020T) and MLi-2 dataset | https://www.ebi.ac.uk/emdb/EMD-45592 | Electron Microscopy Data Bank, EMD-45592 |
| Chen S, Leschziner AE, Villa E | 2024 | Focused refinement map of WD40:ARM/ANK interface from LRRK2(I2020T) MLi-2 dataset | https://www.ebi.ac.uk/emdb/EMD-45593 | Electron Microscopy Data Bank, EMD-45593 |

*Continued on next page*

*Continued*

| Author(s) | Year | Dataset title | Dataset URL | Database and Identifier |
|---|---|---|---|---|
| Chen S, Basiashvili T, Hutchings J, Murillo MS, Suarez AV, Louro JA, Leschziner AE, Villa E | 2024 | Autoinhibited full-length LRRK2(I2020T) on microtubules with MLi-2 | https://doi.org/10.2210/pdb9cho/pdb | Worldwide Protein Data Bank, 10.2210/pdb9cho/pdb |

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
