## [Editor Report · eLife Assessment]

In this manuscript, Chen et al. used cryo-ET and in vitro reconstituted system to demonstrate that the autoinhibited form of LRRK2 can also assemble into filaments on the microtubule surface, with a new interface involving the N-terminal repeats that were disordered in the previous active-LRRK2 filament structure. The structure obtained in this study is the highest resolution of LRRK2 filaments done by subtomogram averaging, representing a major technical advance compared to the previous paper from the same group. This is an **important** study, especially considering the pharmacological implications of the effect of inhibitors of the protein. The strengths of the data are **convincing**, but the study would be considerably strengthened if the authors explored the physiological significance of the new interfaces and the **incomplete** decoration of microtubules described here.

---

## [Referee Report · Reviewer #1 (Public review)]

Summary:

In this manuscript, Chen et al. use cryo-electron tomography and an in vitro reconstitution system to demonstrate that the autoinhibited form of LRRK2 can assemble into filaments that wrap around microtubules. These filaments are generally shorter and less ordered than the previously characterized active-LRRK2 filaments. The structure reveals a novel interface involving the N-terminal repeats, which were disordered in the earlier active filament structure. Additionally, the autoinhibited filaments exhibit distinct helical parameters compared to the active form.

Strengths:

This study presents the highest-resolution structure of LRRK2 filaments obtained via subtomogram averaging, marking a significant technical advance over the authors' previous work published in Cell. The data are well presented, with high-quality visualizations, and the findings provide meaningful insights into the structural dynamics of LRRK2.

Weaknesses and Suggestions:

The revised manuscript by Chen et al. has fully addressed all of my previous suggestions regarding the rearrangement of the main figures.

---

## [Referee Report · Reviewer #2 (Public review)]

The authors of this paper have done much pioneering work to decipher and understand LRRK2 structure and function and uncover the mechanism by which LRRK2 binds to microtubules and to study the roles that this may play in biology. Their previous data demonstrated that LRRK2 in the active conformation (pathogenic mutation or Type I inhibitor complex) bound to microtubule filaments in an ordered helical arrangement. This they showed induced a "roadblock" in the microtubule impacting vesicular trafficking. The authors have postulated that this is a potentially serious flaw with Type 1 inhibitors and that companies should consider generating Type 2 inhibitors in which the LRRK2 is trapped in the inactive conformation. Indeed the authors have published much data that LRRK2 complexed to Type 2 inhibitors does not seem to associate with microtubules and cause roadblocks in parallel experiments to those undertaken with type 1 inhibitors published above.

In the current study the authors have undertaken an in vitro reconstitution of microtubule bound filaments of LRRK2 in the inactive conformation, which surprisingly revealed that inactive LRRK2 can also interact with microtubules in its auto-inhibited state. The authors' data shows that while the same interphases are seen with both the active LRRK2 and inactive microtubule bound forms of LRRK2, they identified a new interphase that involves the WD40-ARM-ANK- domains that reportedly contributes to the ability of the inactive form of LRRK2 to bind to microtubule filaments. The structures of the inactive LRRK2 complexed to microtubules are of medium resolution and do not allow visualisation of side chains.

This study is extremely well written and the figures incredibly clear and well presented. The finding that LRRK2 in the inactive autoinhibited form can associate with microtubules is an important observation that merits further investigation. This new observation makes an important contribution to the literature and builds upon the pioneering research that this team of researchers has contributed to the LRRK2 fields.

Comments on revised version:

The authors have adequately addressed my questions and those of the other Reviewers in my opinion.

---

## [Referee Report · Reviewer #3 (Public review)]

Summary:

The manuscript by Chen et al examines the structure of the inactive LRRK2 bound to microtubules using cryo-EM tomography. Mutations in this protein have been shown to be linked to Parkinson's Disease. It is already shown that the active-like conformation of LRRK2 binds to the MT lattice, but this investigation shows that full-length LRRk2 can oligomerize on MTs in its autoinhibited state with different helical parameters than were observed with active-like state. The structural studies suggest that the autoinhibited state is less stable on MTs.

Strengths:

The protein of interest is very important biomedically and a novel conformational binding to microtubules in proposed

The authors have addressed my original critique.

---

## [Author Response]

The following is the authors’ response to the original reviews

**Public Reviews:**

**Reviewer #1 (Public review):**
Summary:In this manuscript, Chen et al. used cryo-ET and in vitro reconstituted system to demonstrate that the autoinhibited form of LRRK2 can also assemble into filaments that wrap around the microtubule, although the filaments are typically shorter and less regular compared to the previously reported active-LRRK2 filaments. The structure revealed a new interface involving the N-terminal repeats that were disordered in the previous active-LRRK2 filament structure. The autoinhibited-LRRK2 filament also has different helical parameters compared to the active form.Strengths:The structure obtained in this study is the highest resolution of LRRK2 filaments done by subtomogram averaging, representing a major technical advance compared to the previous Cell paper from the same group. Overall, I think the data are well presented with beautiful graphic rendering, and valuable insights can be gained from this structural study.Weaknesses:(1) There are only three main figures, together with 9 supplemental figures. The authors may consider breaking the currently overwhelming Figures 1 and 3 into smaller figures and moving some of the supplemental figures to the main figure, e.g., Figure S7.(2) The key analysis of this manuscript is to compare the current structure with the previous active-LRRK2 filament structure. Currently, such a comparison is buried in Figure 3H. It should be part of Figure 1.

We thank the reviewer for this suggestion. As suggested, we have rearranged the figures, split Figure 1 and 3 into smaller Figures, and moved the comparison analysis in Figure 3H to the new Figure 1. Specifically, the old Figure 1 is separated into two figures, introducing the model-building process and describing the two symmetric axes. The old Figure 3 is also separated into two small figures, describing the geometric analysis and model comparison, respectively.

**Reviewer #2 (Public review):**
The authors of this paper have done much pioneering work to decipher and understand LRRK2 structure and function, to uncover the mechanism by which LRRK2 binds to microtubules, and to study the roles that this may play in biology. Their previous data demonstrated that LRRK2 in the active conformation (pathogenic mutation or Type I inhibitor complex) bound to microtubule filaments in an ordered helical arrangement. This they showed induced a "roadblock" in the microtubule impacting vesicular trafficking. The authors have postulated that this is a potentially serious flaw with Type 1 inhibitors and that companies should consider generating Type 2 inhibitors in which the LRRK2 is trapped in the inactive conformation. Indeed the authors have published much data that LRRK2 complexed to Type 2 inhibitors does not seem to associate with microtubules and cause roadblocks in parallel experiments to those undertaken with type 1 inhibitors published above.In the current study, the authors have undertaken an in vitro reconstitution of microtubule-bound filaments of LRRK2 in the inactive conformation, which surprisingly revealed that inactive LRRK2 can also interact with microtubules in its auto-inhibited state. The authors' data shows that while the same interphases are seen with both the active LRRK2 and inactive microtubule bound forms of LRRK2, they identified a new interphase that involves the WD40-ARM-ANK- domains that reportedly contributes to the ability of the inactive form of LRRK2 to bind to microtubule filaments. The structures of the inactive LRRK2 complexed to microtubules are of medium resolution and do not allow visualisation of side chains.This study is extremely well-written and the figures are incredibly clear and well-presented. The finding that LRRK2 in the inactive autoinhibited form can be associated with microtubules is an important observation that merits further investigation. This new observation makes an important contribution to the literature and builds upon the pioneering research that this team of researchers has contributed to the LRRK2 fields. However, in my opinion, there is still significant work that could be considered to further investigate this question and understand the physiological significance of this observation.

We thank the reviewer for the positive comments and we agree that more work can be done next to understand the physiological significance of the autoinhibited LRRK2 in cellular environments. We are actively working on understanding how the stability of autoinhibited full-length LRRK2 is regulated, especially how the transfer between autoinhibited and active forms of LRRK2 can happen. Our in situ data (Watabane et al. 2020) indicates that overexpressed hyperactive PD-mutant LRRK2 mainly adopts its active-like conformation in cells. Thus, learning how the state transfer occurs will allow us to target autoinhibited LRRK2 specifically and efficiently in cells and study its structure and function in physiological conditions.

**Reviewer #3 (Public review):**
Summary:The manuscript by Chen et al examines the structure of the inactive LRRK2 bound to microtubules using cryo-EM tomography. Mutations in this protein have been shown to be linked to Parkinson's Disease. It is already shown that the active-like conformation of LRRK2 binds to the MT lattice, but this investigation shows that full-length LRRk2 can oligomerize on MTs in its autoinhibited state with different helical parameters than were observed with the active-like state. The structural studies suggest that the autoinhibited state is less stable on MTs.Strengths:The protein of interest is very important biomedically and a novel conformational binding to microtubules in the proposed.Weaknesses:(1) The structures are all low resolution.

We thank the reviewer for the comments on both the strengths and weaknesses of the manuscript. We agree with the reviewer that higher resolution would provide more information about how LRRK2 interacts with microtubules and oligomerizes in its autoinhibited form. However, with the current resolution, our model-building benefited significantly from the published high-resolution models and the alpha-fold predictions. We used cryo-ET and subtomogram analysis to solve the structure because this filament is less regular than the right-handed active LRRK2 filament, preventing us from using conventional single-particle analysis. As highlighted by reviewer 1, being able to push the resolution to sub-nanometer is an important advance reflecting state-of-the-art subtomogram analysis, especially for a heterogeneous sample. Notably, the microtubule reconstruction reached higher resolution, comparable to our previous single-particle studies on LRRK2-RCKW (Snead and Matyszewski et al.), confirming the data quality.

(2) There are no measurements of the affinity of the various LRRK2 molecules (with and without inhibitors) to microtubules. This should be addressed through biochemical sedimentation assay.

We thank the reviewer for the suggestion and we agree that learning the binding affinity between LRRK2 and microtubules would be informative. We attempted to purify the LRRK2 with mutants on the WD40:ARM/ANK interface we identified in the manuscript.. Unfortunately, either LRRK2 or LRRK2^I2020T^ with N-terminal mutants (R521A/F573A/E854K), the yield and purity of the final samples are significantly worse than our routine LRRK2 prep. Our chromatography and gel electrophoresis results indicate that proteins are degrading during purification.

**Author response image 1. sa4fig1:** 

While we have attached the results here, and it would be interesting to investigate why N-terminal mutations destabilize LRRK2, we anticipate that significant efforts would be required for further experiments, which we respectfully consider outside of the scope of this manuscript.

**Recommendations for the authors:**

**Reviewer #1 (Recommendations for the authors):**
(1) In Figure S9, the graphic definition of "chain length" in panel A is misleading. The authors can simply note in the figure legend that "chain length is the number of asymmetric units in a continuous chain".

We thank the reviewer for the suggestion. The updated figure and legend have incorporated the changes.

(2) In Figure S7B, the conformation changes of the 'G-loop' and the 'DYG' motifs are not so convincing at the current resolution.

We thank the reviewer for pointing it out. We agree that our model resolution is not high enough to support the unbiased observation of the conformation changes of the key kinase motifs. In the revised manuscript, we avoided emphasizing the comparison between the two models. Instead, we state that for both the MLi-2 bound map and the GZD-824 bound map, the corresponding published high-resolution models fit into each kinase map, but the MLi-2 bound model doesn’t fit as well in the GZD-824 bound map, with a correlation value dropped from 0.44 to 0.4, supporting our statement that “full-length LRRK2 bound to microtubules is in its autoinhibited state in our reconstituted system”.

**Reviewer #2 (Recommendations for the authors):**
(1) Are there any cellular experiments that could be done to demonstrate that inactive LRRK2 associates with microtubules in cells?

We thank the reviewer for pointing out this direction for future studies. We are studying the physiological significance of the autoinhibited LRRK2 in cells, but haven’t yet been successful at demonstrating physiological binding to microtubules. Further, as noted in our response to reviewer #3, we are also actively working on understanding how the stability of autoinhibited full-length LRRK2 is regulated, especially how the transfer between autoinhibited and active forms of LRRK2 can happen. Our in situ data (Watabane et al. 2020) indicates that hyperactive PD-mutant overexpressed LRRK2 mainly adopts its active-like conformation in cells. Thus, learning how the state transfer occurs will allow us to target autoinhibited LRRK2 specifically and efficiently in cells and study its structure and function in physiological conditions.

(2) Previous work that the authors and others have undertaken has suggested that only LRRK2 in its active conformation can associate with microtubule filaments and the authors have shown that this leads to a roadblock in vesicular transport only when LRRK2 is complexed with Type 1 but not Type 2 inhibitors. There seems to be some discrepancy here that is not addressed in the paper as based on the current results one would also expect LRRK2 bound to Type 2 inhibitors to induce roadblocks in microtubule filaments. How can this be explained?

We thank the reviewer for raising this important question. Taking all of our published data together, we believe that LRRK2 can introduce roadblocks with Type 1 inhibitor bound in the active-like conformation, where N-terminus LRRK2 domains are flexible and don’t block the kinase active site. In other words, full-length LRRK2 can form roadblocks when it behaves more like the truncated LRRK2^RCKW^ variant. The autoinhibited LRRK2 forms shorter and less stable oligomers on microtubules, making it harder to block transport. Consistent with this, our in situ LRRK2-microtubule structure was observed in cells where LRRK2 is in an active-like conformation, and the LRRK2 N-terminus appeared to be flexible and away from the microtubule when forming right-handed filaments.

(3) Does the finding that inactive LRRK2 only binds to microtubules as a short filament, explain the differences between the inactive and active forms of LRRK2 binding to microtubules and causing roadblocks?

We thank the reviewer for discussing this point with us and asking the question. As we replied in the previous comment, the reviewer’s conclusion explains how the roadblock phenomenon occurs only under certain circumstances. We expanded our discussion to add the following and address the question:

“Notably, we previously demonstrated that active‐like LRRK2, when bound to a Type I inhibitor, can form roadblocks that impair vesicular transport. Since autoinhibited LRRK2 assembles into shorter, less stable oligomers on microtubules, we anticipate it will exert reduced road‐blocking effects in cells, regardless of the inhibitor bound.”

(4) Could the authors undertake further characterization of the new WD40-ARM-ANK interphase that they have identified? Is this important for the binding of the autoinhibited mutant? Could mutants be made in this interphase to see if this prevents the autoinhibited but not the active conformation of LRRK2 binding to microtubules?

We thank the reviewer for the comment. As mentioned in our response to Reviewer #2, public comment #2, we attempted to purify the LRRK2 with mutants on the WD40:ARM/ANK interface we identified in the manuscript multiple times. Unfortunately, either LRRK2 or LRRK2^I2020T^ with N-terminal mutants (R521A/F573A/E854K), the yield and purity of the final samples are significantly worse than our routine LRRK2 prep. Our chromatography and gel electrophoresis results indicate that proteins are degrading during purification.

(5) The authors identify several disease-relevant missense mutations that appear to lie within the novel interphase that the authors have characterised in this study. Although this is discussed in the Discussion, some experimental data demonstrating how these missense mutations impact the ability of inactive LRRK2 to bind to microtubule filaments in the presence or absence of Type 1 and Type 2 compounds could provide further experimental data that emphasises the physiological importance of the results presented in this study.

We thank the reviewer for discussing this interesting direction. The disease-relevant missense mutations can have a direct or indirect impact on the binding of autoinhibited LRRK2 to microtubules, and we agree that it would be interesting to test it out in the future. However, we anticipate that significant effort would be required for further experiments. Alas, our funding for this project ended suddenly and we want to report our results to the community.

(6) For the data that is shown in Figure 1, could the authors explain how this differs from results in previous papers of the authors showing that the active form of LRRK2 binds microtubules? How does the binding observed here differ from that observed in the previous studies? To a non-specialist reader, the data looks fairly like what has previously been reported.

We thank the reviewer for asking the question. As mentioned in the response to the public review, the detailed comparison between the data and the previous papers is described in Figure 3, and we agree that it is helpful to incorporate this information in Figure 1. In the revised manuscript, we have incorporated the comparison panel in Figure 1.

(7) The finding that the autoinhibited LRRK2 forms short and sparse oligomers on microtubules raises the question of how physiological this observation is. Having some data that suggests that this is physiologically relevant would boost the impact of this study.

We agree with the reviewer on this comment. As discussed in the response to the first comment from the reviewer, we have not been able to assess the physiological relevance of LRRK2 binding to microtubules in either active or inactive state, but continue to pursue this line of research. We are aware and regret that this lessens the impact of this work.

(8) For the more general reader the authors could potentially better highlight why the key finding in this paper is important.

We thank the reviewer for the suggestion. To further address the significance of the key findings, especially how it can open up more possibilities for inhibitor-based drug development, we expand our discussion section to include the following:

“Understanding how Type I and Type II inhibitors’ binding to LRRK2 affects its mechanism is vital to the design of inhibitor-based PD drug development strategies. Our findings revealed that different LRRK2 kinase inhibitors bind to autoinhibited LRRK2 similarly either in solution or on microtubules. Furthermore, the observation of autoinhibited LRRK2 forming short, less stable oligomers on microtubules opens new possibilities to inhibit LRRK2 activity in PD patients. A Type I inhibitor specifically targeting autoinhibited LRRK2 may alleviate the effect of LRRK2 roadblocks on microtubules. Alternatively, a promising strategy of LRRK2 inhibitor design can focus on the stabilization of allosteric N-terminus blocking on the kinase domain, which favors the formation of autoinhibited LRRK2 oligomers on microtubules and causes fewer side effects.”

**Reviewer #3 (Recommendations for the authors):**
In the third paragraph of the introduction, expand on whether type-1 inhibitors which "capture kinases in a closed, "active-like" conformation still inhibit the kinase activity.

We thank the reviewer for the request to expand this paragraph. We added the following explanation for better understanding in the third paragraph:

“Type-I inhibitors bind to the ATP binding site and target the kinase in its ‘active-like' conformation, inhibiting its kinase activity.”